# Responsiveness and Minimal Clinically Important Difference of the Five Times Sit-to-Stand Test in Patients with Stroke

**DOI:** 10.3390/ijerph18052314

**Published:** 2021-02-26

**Authors:** Rodrigo Martín-San Agustín, Mª José Crisostomo, Mª Piedad Sánchez-Martínez, Francesc Medina-Mirapeix

**Affiliations:** 1Department of Physical Therapy, University of Valencia, 46010 Valencia, Spain; rodrigo.martin@uv.es; 2Department of Rehabilitation, Jerez Hospital, 11407 Jerez de la Frontera, Spain; mjcriso@gmail.com; 3Department of Physical Therapy, University of Murcia, 30100 Murcia, Spain; mirapeix@um.es

**Keywords:** 5STS, stroke, MCID, responsiveness, stages, severity level, gait speed, FAC

## Abstract

This study aimed to analyze the responsiveness of the 5STS test among stroke patients and to estimate the MCIDs (minimal clinically important differences) for different severity levels of community ambulation and stages of recovery. The 5STS and comparator instruments (gait speed and Functional Ambulatory Category (FAC)) were evaluated at baseline. These measures were repeated at 4 (Stage 1) and 8 weeks (Stage 2), together with the Global Rating of Change (GROC). The MCIDs were calculated with two anchor-based methods using the GROC as the external criterion. Responsiveness to change for the 5STS was estimated analyzing the correlation with changes in the two comparator instruments and their capacity to discriminate improvement. For the 5STS test, while the MCIDs of the limited community ambulators were similar in the two stages (around 3 s), those of the household ambulators decreased from 1.9 s to 0.72 s. Spearman’s rho coefficients showed an acceptable correlation between changes in 5STS and changes for both the FAC and gait speed changes in both stages of recovery. Our study revealed that the 5STS is responsive to functional changes in patients with stroke and that their degree of severity and stage of recovery influence the MCID values of the 5STS.

## 1. Introduction

Many patients have difficulties with sit-to-stand (STS) tasks after stroke, to the extent that improving the performance of this functional task is a common goal for rehabilitation teams [1]. Although multiple variations of the STS task have been adapted as functional tests, expert consensus has identified the five-repetition sit-to-stand test (5STS), which measures the time taken to stand five times from a sitting position as rapidly as possible, as the most suitable test for assessing and monitoring the status of stroke patients [1]. Although time in the 5STS is only one of many aspects of STS tasks that can be measured, this test is gaining increased recognition as an important variable because it has implications on disabilities and falls in stroke, and, consequently, it is a widely used outcome measure [2,3,4]. 

The reliability and validity of the 5STS test have been described in patients with stroke [5]. The metric property “responsiveness”, which is the ability of a measure to detect real changes over time [6], has received less attention in spite of its relevance for outcome measures. Similarly, the minimal clinically important difference (MCID), which is the smallest change that is meaningful to patients [7], has not yet been determined for this test. This paper focuses on these voids and provides MCIDs which may be used by clinicians and researchers. Clinicians can use this information to interpret the relevance of changes observed in a patient; whereas researchers can use it to define the boundary between change or no change among two groups (e.g., those treated with different interventions) and for the calculation of sample sizes [8]. Furthermore, these parameters may be an opportunity to determine responsiveness compared to other relevant outcome measures [9].

It has been suggested that the MCID is not a fixed value, but rather it is influenced by the selected calculation method, baseline severity, or contextual-factors [10,11]. As the concept itself of MCID and the methods of establishing it imply that the patients are those who interpret what change is important or meaningful for them [11], some researchers have suggested that the MCID may be affected by determinant contextual-factors, such as severity, stage of recovery, or time since the last measurement [12,13]. Thus, it was suggested that “to gain a clear picture of the MCIDs for different stroke outcome measures, the MCID will need to be estimated for different stages of recovery and levels of severity”. These authors also suggested that there is an absence of these analyses in many stroke performance measures (e.g., gait speed, step length or balance), and therefore proposed that this approach be initiated with gait speed measures. Despite this proposal, to our knowledge there is still a lack of research to quantify these parameters in the 5STS test as well as in gait speed and other measures. 

The main aims of this study were to analyze the responsiveness of the 5STS test for stroke patients and to estimate the MCIDs of this test for different severity levels and stages while undergoing a physical rehabilitation program. Concerning responsiveness, we hypothesized that the 5STS time would decrease following a physical rehabilitation program and that this change (H1) would correlate significantly to corresponding changes in two comparator measures, walking ability and gait speed, and (H2) it would be possible to discriminate participants with change or no change in these two functional comparator measures. A secondary objective was to estimate the MCIDs of gait speed for different severity levels and stages. 

## 2. Materials and Methods

### 2.1. Study Design and Participants

A prospective study was conducted with a two-month follow-up of patients undergoing a physical rehabilitation program. Measurements were made within two days of admission (T0) and repeated at 4 (T1), and 8 (T2) weeks (Figure 1). 

Participants were prospectively recruited and screened during 2016 and 2018 among these patients admitted to participate in a physical rehabilitation program by the Rehabilitation Service of the Jerez Hospital (Spain) after receiving acute care for a stroke at that hospital or nearby. The inclusion criteria were: (1) older than 30 years, (2) undergoing outpatient physical rehabilitation program after a first-time stroke, and (3) enrollment within four months post stroke. The exclusion criteria were (1) unable to walk independently before the stroke, (2) able at baseline of each stage to walk everywhere independently, or (3) unsuccessful completion of the screening portions of the orientation, language, memory, and reasoning/judgment subscales of the Cognistat [14]. All study patients provided written informed consent. The study protocol was approved by the Jerez Hospital’s ethical committee (approval number: EST-42/16).

### 2.2. Intervention

The physical rehabilitation program was individualized and personalized, adapting to the patient’s evolution, including outpatient education, based on a stroke workshop to train their caregivers, and exercise training. This training included exercises at home, exercise training, and treatment of muscle tone, strength, balance, coordination, activities of daily living, sensory stimulation, paretic muscle stimulation, mirror therapy or gait reeducation.

### 2.3. Measures

At T0, demographic (age, gender) and clinical variables (ischemic or hemorrhagic stroke and affected side) were initially collected from medical records. Performance measures (5STS and gait speed) and comparator instruments were also measured. At T1 and T2 patients were assessed again, together with the Global Rating of Change (GROC). Furthermore, community ambulation categories (household ambulators, limited community ambulators, and community ambulators) were selected to enable the subgroup analysis by levels of severity. These categories were determined by gait speed of <0.4 m/s (household ambulators), between >0.4 m/s and <0.8 m/s (limited community ambulators) and gait speed > 0.8 m/s (unlimited community ambulators) at the baseline of each stage [15].

#### 2.3.1. The 5STS and Gait Speed

The 5STS was measured as the time taken to complete five repetitions of the sit-to-stand task. All sit-to-stand tasks were performed using a chair without an armrest, with a height of 43 cm and a depth of 47.5 cm. Timing began when the patient’s back left the backrest and stopped once the back touched the backrest for the fifth time [16]. Participants unable to complete five repetitions within 1 min were given a score of 60 s, in line with previous research [17].

Gait speed was measured using the 4 m gait speed test (4MGS) and reported in m/s. For the 4MGS test, tape was used to mark out 4 m on a flat unobstructed course within a clinical assessment room. Prior to starting the test, the patients viewed a demonstration of the walk but did not perform a practice walk themselves. Subjects were asked to complete the 4-m walk at their “most comfortable speed” and a stopwatch recorded the time [18]. Timing began after an opportunity for acceleration and was stopped when the patient’s first foot completely crossed the 4 m line. Subjects performed two trials, with the faster time recorded [19]. Participants unable to complete gait speed test were given a score of 0 m/s [20].

#### 2.3.2. Global Rating of Change

The GROC asks participants to rate their overall recovery of their walking ability since last measurement on a 15-point ordinal scale where −7 indicates a very great deal worse, +7 indicates a very great deal better, and 0 indicates no change. We selected this measure because the COSMIN group recommended that patients should be the ones to decide what is important [6].

#### 2.3.3. Comparator Instruments for Responsiveness

Since the 5STS test is a performance-based measure, we also used two performance measures as comparators: the Functional Ambulatory Category (FAC) for the walking ability and the 4MGS for gait speed. The FAC is a common clinical gait assessment scale that distinguishes 6 levels of walking ability on the basis of the amount of physical support required [21]. Briefly, the levels are: 0, meaning that the patient cannot walk at all or needs the help of 2 therapists; 1, the patient requires continuous manual contact to support body weight; 2, the patient requires intermittent or continuous light touch to assist balance; 3, patients who can ambulate on a level surface without manual contact of another person but requires standby guarding of one person; 4, the patient can ambulate independently on a level surface but requires supervision; and 5, the patient can walk everywhere independently. This scale has showed excellent reliability, good concurrent validity with gait speed and predictive validity with community ambulation, and good responsiveness (SRM > 0.8 after physical rehabilitation program of two weeks) [21]. As recommended by Bohannon et al. [9], we used the MCIDs calculated in this study for gait speed as an opportunity to determine responsiveness of the 5STS. In addition, we also used the magnitude in m/s as a reference measure.

### 2.4. Statistical Analysis

Patients’ characteristics at baseline were summarized. Categorical variables were expressed as counts (%), continuous demographic characteristic variables were used by calculating the mean (SD) and stroke characteristics are presented as the median (IQR). The change in scores was expressed as the median (IQR) of the 5STS test and gait speed in each stage of recovery, obtained for all participants and by walking categories. All analyses were performed using MedCalc Statistical Software (MedCalc Software, Mariakerke, Belgium).

This study followed the recommendation of the COSMIN group [6] for determining MCIDs and responsiveness. The MCIDs of the 5STS and gait speed were calculated with two anchor-based methods (the Receiver Operator Characteristics (ROC) method and the within-patients method) [11], using the GROC as the external criterion. GROC scores were dichotomized to define an anchor value able to reflect tangible and marked clinically meaningful improvements in the patient’s condition. Based on the previous literature [14], we selected a GRC change > 5 (i.e., feeling better to feeling a great improvement) to represent an important change and a score ≤ 5 to represent unimportant changes or no change. For the ROC method, the MCID was determined based on a cut-off point that maximized sensitivity and specificity to distinguish between important and unimportant groups. For the within-patients method, the MCID was estimated as the median of changes in scores within the group who self-reported important changes. This method is an anchor-based MCID distribution, by establishing the GROC as anchor and selecting the MCID between the distribution (median) among those patients who improved. In addition, in order to select the best MCID of the two methods, logistic regression was used to identify the odds ratio (OR) of demonstrating a clinical improvement on the GROC, based on having a 5SST score above or below the MCID calculated by each method. The effect of potential confounding covariates was also determined. We selected the MICD with a higher OR.

Responsiveness was evaluated based on the hypothesis described in the introduction section, as recommended by the COSMIN protocol [6]. To test H1, the Spearman rank correlation was used to assess the association between a change in scores in the 5STS, and the change in scores for the FAC and 4MGS. The correlation was classified as strong (Spearman’s rho > 0.70), acceptable (Spearman’s rho = 0.30–0.70), and weak (Spearman’s rho < 0.30) [22]. To test H2, we used ROC curves of the change in scores for the 5STS test with both the FAC and the gait speed change scores dichotomized as improved (≥1 point in the FAC and ≥MCIDs in the gait speed) and non-improved (≤0 and <MICDs, respectively). Thus, the area under curve (AUC) of these ROC curves was calculated, which represents the probability that the measure of correctly classifying patients has either improved from not-improved) [23]. An AUC > 0.700 was used as a generic benchmark to consider its discriminant validity as acceptable [24]. 

## 3. Results

### 3.1. Patients’ Characteristics

In total, 123 stroke patients were screened for inclusion. Figure 1 shows the reasons 12 patients were excluded. At baseline of Stage 1, the age of the 111 patients ranged from 30 to 87 years (mean = 68.3 years, SD = 12.2). Most of the patients (n = 75; 68.2%) were household ambulators. Limited community ambulators were younger (63.9 years versus 70.6 years) and had better performance on the 5STS and higher gait speed than household ambulators (Table 1). Of the 111 patients who began the study at stage 1, 108 finally completed the study. Reasons for dropouts are shown in Figure 2. In Stage 2, 93 patients were initially included, of which 88 completed the study (Figure 2).

Table 2 shows baseline 5STS and gait speed scores of those subjects who completed each stage and Figure 3 shows absolute change score between baseline and at the end of each stage. Limited community ambulators improved (i.e., reduced) a median of 1.81 s in Stage 1 and 1.52 s in Stage 2. In contrast, at least 50% of the patients who were household ambulators did not improve. Regarding gait speed, the limited community ambulators improved (i.e., increased) 0.10 m/s in the two stages. The household ambulators increased their gait speed, especially for Stage 1 where at least 25% of them improved from not being able to do the test to having a speed of 0.19 m/s.

### 3.2. Minimal Clinically Important Difference

Table 3 shows that the MCIDs of the 5STS and gait speed differed according to the method used in almost all patient groups and stages, and how all the selected MCIDs were those based on the ROC method (i.e., they met our previously established selection criteria: showing higher ORs).

For the 5STS test, whereas the selected MCIDs of the limited community ambulators were similar in the two stages (around 3 s), those of the household ambulators decreased from 1.9 s to 0.72 s. In each stage, the MCIDs of household ambulators and limited community ambulators showed differences, especially in the second stage (0.72 s vs. 3.09 s, respectively). Regarding gait speed, both household ambulators and limited community ambulator patients showed similar MCIDs in stage 1 (around 0.20 m/s). Nevertheless, the two groups decreased their values in the second stage (e.g., limited community ambulators decreased from 0.21 m/s to 0.11 m/s).

### 3.3. Responsiveness

Table 4 shows the correlations and AUC between the changes in the 5STS and changes in FAC and gait speed by stage. Spearman’s rho coefficients showed acceptable correlations for most relationships between changes in 5STS and changes for the FAC and gait speed, both in Stage 1 (ranging from 0.302 to 0.566) and in Stage 2 (ranging from 0.301 to 0.448).

The ability for changes in the 5STS test to discriminate patients who improved or failed to improve their scores for both the FAC and gait speed measures was generally acceptable (AUCs around or greater than 0.70). The ability for changes in the 5STS to discriminate improvements in patients’ gait speed was slightly better (AUCs’ range between 0.71–0.75) compared to the FAC (AUCs’ range between 0.67–0.68) during Stage 1, and again during Stage 2 (0.70–0.74 vs. 0.66–0.76, respectively.

## 4. Discussion

This study demonstrated that the 5STS test is responsive to physical rehabilitation program in stroke patients. Considering patients in both stages of stroke recovery, changes in the 5STS for patients who were household ambulators, limited community ambulators, and of the whole sample correlated significantly with changes in two comparators: walking ability and gait speed. Moreover, we found that these changes in scores were also able to discriminate between meaningful changes and no changes in these two comparators, especially in relation to gait speed. The estimated MCIDs of 5STS showed variations between the two severity groups in both stages, however, only the MICDs of the household ambulators showed variations over stages of recovery. The MICDs of gait speed showed variations between the two severity groups only in patients in the second stage of recovery. Although their MCIDs showed variations over stages, they also decreased more for the household ambulators. 

Our hypotheses regarding responsiveness were verified. The changes in the 5STS scores of both household ambulators and limited community ambulators, correlated moderately with the FAC and the gait speed, which are well recognized prognostic markers of community ambulation [21,25,26]. Therefore, these correlations suggest that the 5STS may also be considered a prognostic marker. 

Although previous cross-sectional studies found higher correlations between the 5STS and gait speed in other neurological diseases [27], in our study we expected moderate correlations between changes in scores. This is due to the fact that the 5STS test is a global measure which is dependent on sensation, speed, balance, psychological status, and strength [3]. However, gait speed or walking ability may require a lesser magnitude of these aspects, and additionally other aspects [28]. Thus, some participants could have experienced changes in gait speed or walking ability; however, these were not reflected in a similar change in the 5STS time, or vice versa. In addition, as expected, correlations and AUCs of the FAC were lower than those of gait speed. This was expected because walking ability was measured on an ordinal scale (i.e., FAC), and consequently the 5STS time and other functional tasks (e.g., gait speed) can provide slight improvements with treatment irrespective of any improvement in walking ability.

To the best of our knowledge, no previous studies have established MCIDs for the 5STS among stroke patients. This study used the GROC as an anchor criterion to establish all the MCIDs. We found two relevant findings. First, the MCIDs of the household ambulators (i.e., the more severe patients) were slightly lower than those of the limited community ambulators in both stages of recovery. A possible reason for this, is that patients that are more severe tend to consider smaller amounts of change as important [29]. Moreover, only the MCIDs of household decreased over stages of recovery. One possible explanation is that while any improvement in the patient’s condition is clinically important for all patients—more and less severe—at the beginning of a physical rehabilitation program, this perception might change for some patients in later stages of recovery in which lesser improvements are often achieved. In contrast, although patients who are less severe may maintain the same initial perception, patients with worse baseline values (severe patients) might reduce their thresholds of clinically important change. 

Regarding gait speed, our study estimated that patients in the first stage of recovery had a MCID of 0.19m/s for all patients. This value was very similar to the value found by Fulk et al. [14] (0.18 m/s), which also used the GROC of the patient to establish MCIDs in stroke patients. In addition, our study shows that MCIDs fluctuate over time, with more conservative values in the early stages of a physical rehabilitation program and higher differences between severity groups as it progresses. We were unable to compare these results with previous studies because previous studies seldom considered variations in severity level or stages of recovery. Nevertheless, we agree with previous studies that establish that the MCID should not be considered as a fixed value [10], but rather the exact value for the MCIC should be determined considering aspects such as the initial scores, the target population, and the method used to assess the MCID.

### Strengths and Limitations of the Study

This study had several strengths. First, this study followed the considerations established in the literature for the analysis of MCIDs by regarding severity levels and stages of recovery. Second, we used two methods to calculate the MCID-ROC method and within-patients method—and additionally we selected the MCID which best discriminated between patients who have improved for each variable or not, severity level and stage. For the two methods and the two tests, we used an anchor based on the patient’s perception, as recommended [11]. Some authors who estimated MCIDs for gait speed did not include this same anchor. For example, Tilson et al. 2010 [12] used the modified Rankin Scale as an anchor, which is a global index of disability broadly, for calculating the MCID of the gait speed in household ambulators after stroke. Similarly, Perera et al. 2006 [30] used the SF-36 as an anchor in limited community ambulators. Third, we used two functional measures (walking ability and gait speed) as comparators for analyzing the responsiveness of the 5STS, as well as using two methods of analysis (correlations and AUC).

Despite these novel findings, this study was also subject to some limitations. First, many household patients were unable to complete the tests at the baseline of each stage and therefore these values were imputed. These participants provided outliers and asymmetric distributions of change either because many of them continued to be unable to perform the test at the end of a stage (with null change) or they had marked changes when they were able to do the test (e.g., when a patient who was initially unable improved, achieving a 5STS of 25 s this was considered a marked change [60 s − 25 s = 35 s]). Consequently, means and effect sizes were not calculated to describe changes. Nevertheless, medians, RIQ, and percentage of change were sufficiently descriptive. Second, we classified the patients who were unable to do the 4MGS within the household group. We include these non-ambulator patients in this group since they could progress thanks to physical rehabilitation program. However, in our opinion, further research is necessary to establish differences in metric properties between non-ambulators and household ambulators.

## 5. Conclusions

Our study showed that the 5STS is responsive to functional changes in patients with stroke and that their degree of severity and stage of recovery may influence the MCID values of the 5STS over time. Patients who were more severe presented lower MICDs in both stages of recovery than those who were less severe. The MCIDs for gait speed also showed that these were influenced by stages of recovery, with more conservative values in the initial stages. We confirmed that clinicians and researchers need to consider stages of recovery and levels of severity when using MCIDs for stroke patients. 

## Figures and Tables

**Figure 1 ijerph-18-02314-f001:**
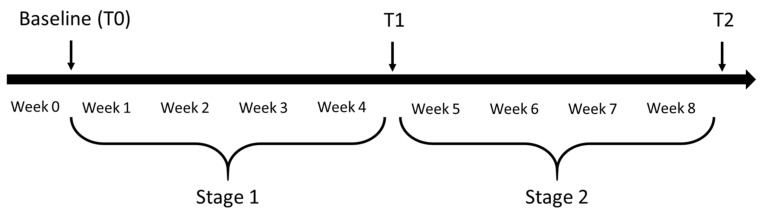
Study timeline.

**Figure 2 ijerph-18-02314-f002:**
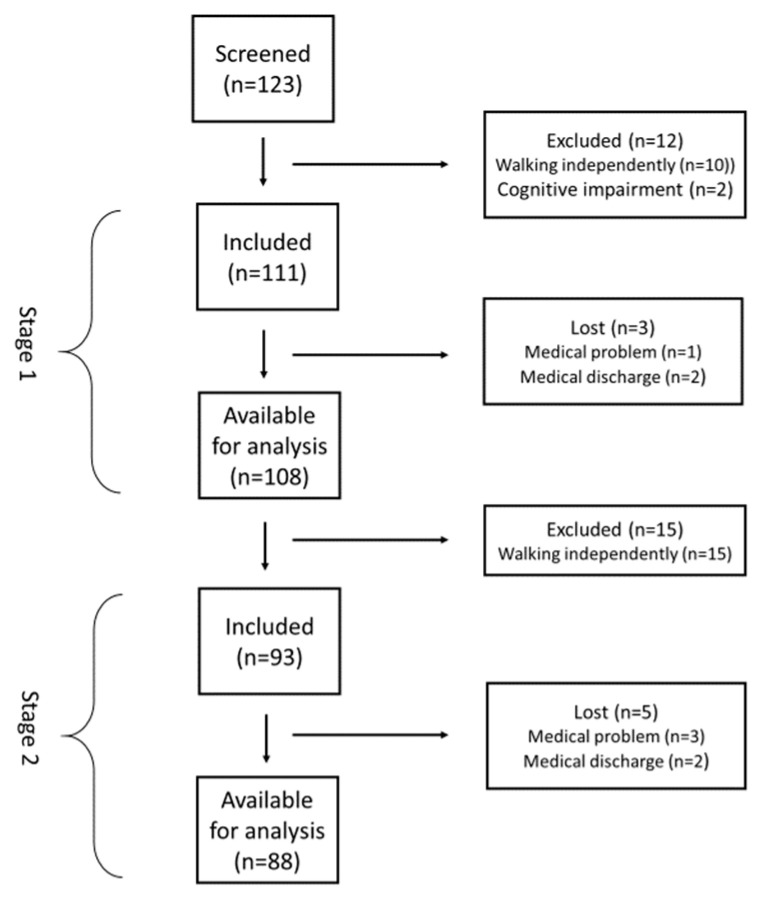
Flowchart of patients who met inclusion/exclusion criteria for the study population.

**Figure 3 ijerph-18-02314-f003:**
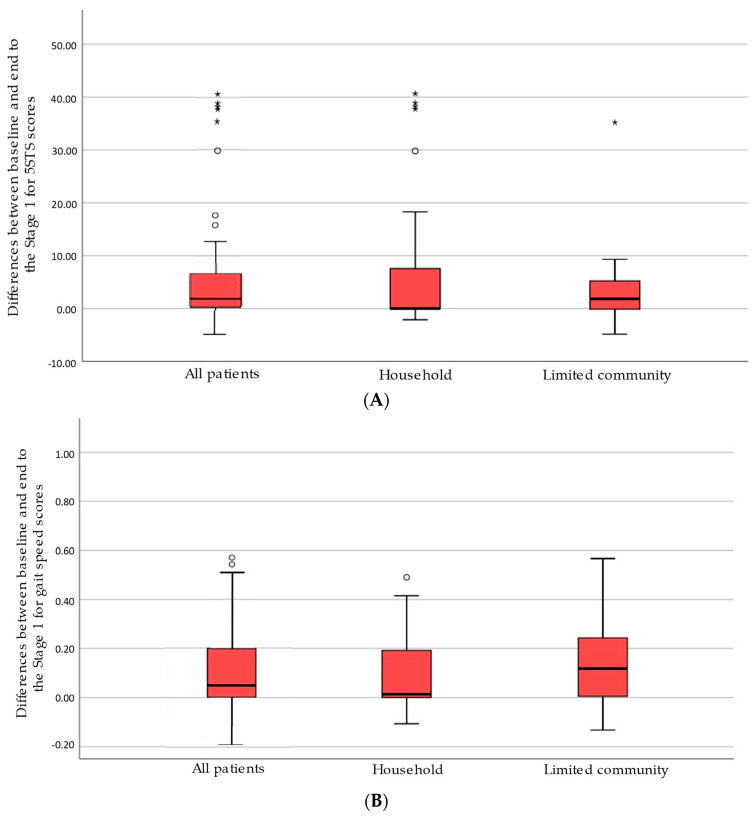
Boxplots of the differences between baseline and at the end of each stage of 5STS and gait speed scores in the Stage 1 (**A**,**B**) and in the Stage 2 (**C**,**D**) in all patients and by walking category.

**Table 1 ijerph-18-02314-t001:** Baseline demographics and overall stroke patient characteristics in total and separated by walking category. SD = standard deviation; PT = physical therapy; FAC = Functional Ambulation Category; IQR = interquartile range; 5STS = five-repetition sit-to-stand; 4MGS = 4 m gait speed.

	All Patients (n = 111)	Walking Category		
		Household (n = 76)	Limited Community (n = 31)	Subgroup Differences *p* Value
Age. years; mean (SD)	68.3 (12.1)	70.6 (11.1)	63.9 (13.3)	0.02
Gender (male)	60 (54%)	39 (35%)	18 (16%)	0.61
Side affected (right)	59 (53%)	37 (33%)	19 (17%)	0.29
Type of stroke (ischemic)	91 (82%)	62 (56%)	26 (23%)	0.90
Time from stroke to admission to outpatient PT. days; mean (SD)	51.8 (31.5)	53.8 (31.9)	50.3 (31.2)	0.87
FAC. score 0–5; median (IQR)	2 (2)	1 (3)	4 (1)	0.01
Patients unable to do 5STS/4MGS	44/44	42/44	2/0	-
5STS. s; median (IQR)	29.75 (43.80)	60 (35.80)	15.96 (6.39)	0.01
Gait speed. m/s; median (IQR)	0.27 (0.43)	0 (0.27)	0.59 (0.26)	0.01

**Table 2 ijerph-18-02314-t002:** Medians (interquartile range) of the 5STS time and gait speed at baseline in all patients and by walking category.

	Baseline	
	At Stage 1 n_1_ = 108; n_2_ = 66; n_3_ = 42	At Stage 2 n_1_ = 88; n_2_ = 62; n_3_ = 26
**5STS (s)**		
All patients (n_1_)	29.97 (43.68)	23.5 (44.53)
Household (n_2_)	60 (34.1)	60 (38.37)
Limited community (n_3_)	15.96 (6.39)	16.4 (8.87)
**Gait speed (m/s)**		
All patients (n_1_)	0.27 (0.45)	0.37 (0.49)
Household (n_2_)	0 (0.26)	0.15 (0.27)
Limited community (n_3_)	0.59 (0.26)	0.52 (0.17)

5STS = five-repetition sit-to-stand.

**Table 3 ijerph-18-02314-t003:** The minimum clinically important difference (MCIDs) for the performance measures using the global rating of change scale by stage.

	Stage1			Stage 2		
	ROC Method		Within Patients	ROC Method		Within Patients
	AUC	MCIDs (Associated ORs ^a^)	MCIDs (Associated ORs ^a^)	AUC	MCIDs (Associated ORs ^a^)	MCIDs (Associated ORs ^a^)
**5STS (s)**						
All patients	0.71	1.18 ^†^ (2.44 *)	2.81 (1.64 *)	0.73	0.76 ^†^ (3.87 *)	2.02 (3.31 *)
Household	0.72	1.90 ^†^ (4.13 *)	5.1 (2.46 *)	0.76	0.72 ^†^ (6.67 *)	1.94 (5.67 *)
Limited community	0.70	2.92 ^†^ (0.70 *)	1.61 (0.68 *)	0.70	3.09 ^†^ (4.96 *)	2.09 (1.07 *)
**Gait speed (m/s)**						
All patients	0.71	0.19 ^†^ (3.71 *)	0.11 (2.67 *)	0.71	0.09 ^†^ (2.88 *)	0.09 (2.88 *)
Household	0.74	0.19 ^†^ (6.19 *)	0.10 (3.71 *)	0.70	0.04 ^†^ (8.17 *)	0.07 (2.76 *)
Limited community	0.70	0.21 ^†^ (1.83 *)	0.17 (1.43 *)	0.73	0.11 ^†^ (3.71 *)	0.12 (2.63 *)

ROC = Receiver Operator Characteristics; AUC = area under curve; 5STS = five-repetition sit-to-stand; ORs = odds ratio. ^a^ Associated Odd Ratios to MCIDs. ^†^ MCID with higher odd ratio. * *p* < 0.05.

**Table 4 ijerph-18-02314-t004:** Responsiveness statistics for the STS by stage.

	Stage 1		Stage 2	
	Spearman’s Rho	AUC	Spearman’s Rho	AUC
**FAC**				
All patients	0.307 *	0.68	0.357 *	0.71
Household	0.332 *	0.68	0.308 *	0.66
Limited community	0.302 *	0.67	0.448 *	0.76
**Gait speed (m/s)**				
All patients	0.492 *	0.73	0.376 *	0.72
Household	0.566 *	0.75	0.431 *	0.74
Limited community	0.396 *	0.71	0.301 *	0.70

CI = confidence interval; AUC = area under curve; 5STS = five-repetition sit-to-stand. * Significant differences set at *p* < 0.05.

## Data Availability

The data presented in this study are available on request from the corresponding author.

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
