# Peer review of "Responsiveness and Minimal Clinically Important Difference of the Five Times Sit-to-Stand Test in Patients with Stroke"

_ijerph, 2021, doi:10.3390/ijerph18052314_

Round 1

Reviewer 1 Report

This study analyzed the responsiveness of the 5STS test among stroke patients and to estimate the MCIDs for different severity levels of community ambulation and stages of recovery. They revealed that the 5STS is responsive to functional changes in patients with stroke and that their degree of severity and stage of recovery influence the MCID values of the 5STS. I do have some comments as listed below in the order noted.

First, in longitudinal designs, anchor-based methods might better reflect clinical importance, while distribution-based methods might be better suited for cross-sectional designs. Please provide a distribution-based method to determine the MCID for the present study.

Second, please added p values to compare baseline demographics and overall stroke patient characteristics separated by walking category in Table 1.

Overall, please also provide boxplots of both 5STS time and gait speed absolute change score between baseline and at the end of each stage in all patients and by walking category.

Author Response

  • Comment #1: This study analyzed the responsiveness of the 5STS test among stroke patients and to estimate the MCIDs for different severity levels of community ambulation and stages of recovery. They revealed that the 5STS is responsive to functional changes in patients with stroke and that their degree of severity and stage of recovery influence the MCID values of the 5STS. I do have some comments as listed below in the order noted.

Response: We want to thank the reviewer 1 the consideration for the manuscript.  

  • Comment #2: First, in longitudinal designs, anchor-based methods might better reflect clinical importance, while distribution-based methods might be better suited for cross-sectional designs. Please provide a distribution-based method to determine the MCID for the present study.

Response: Thank you very much for your comment. Yes, we agree. This is a point discussed in the literature. There seems to be a common consensus that to establish MCIDs, it is appropriate to use several methods. Regarding the method, whether anchor-based or distribution-based method, there is more controversy. The most recent articles that discuss this, propose the combination of both methods, calling this method anchor-based MCID distribution (Devet et al. 2006). As you can see, this combinatorial method is one of those that we have used when we establish an anchor (GROC) and of those patients that improve, we select the MCIDs by their distribution (median). To make this clear, we have indicated it in the text.

  • Comment #3: Second, please added p values to compare baseline demographics and overall stroke patient characteristics separated by walking category in Table 1.

Response: Thank you so much. We have added this information.

  • Comment #4: Overall, please also provide boxplots of both 5STS time and gait speed absolute change score between baseline and at the end of each stage in all patients and by walking category.

Response: Thank you very much for your suggestions. We have incorporated both the 5STS and gait speed boxplots for each phase. In order not to repeat information, we have eliminated the differences between medians from table 2.

Reviewer 2 Report

The aim of this study is to analyze the responsiveness of the 5STS test among stroke patients and to estimate the MCIDs for different severity levels of community ambulation and stages of recovery. The reviewer considers that this study to be clinically important, with appropriate study design and data analysis. However, there are some questions in this study.

As a comment, as the authors also describe in the “Introduction” session, the 5STS test affects the prognosis and movement disorders of stroke, but it is only one of many aspects. It is not possible to predict the prognosis or movement disorders of stroke with the 5STS test alone. Therefore, reviewer believes that combining the results of other performance tests will make the data even more implicative.

The information of stroke patients is poor in this study. The authors should add patients’ background to the “Materials and Methods” session. Particularly, what was the breakdown of the pathosis of 111 stroke patients? (e.g., cerebral infarction, cerebral hemorrhage and subarachnoid hemorrhage). Moreover, how do these differences in pathosis affect 5STS time?

Additionally, the reviewer believes that 5STS time is strongly affected by the differences in sequelae of stroke (e.g., hemiplegia and movement disorders). How do the authors interpret the effects of differences in these sequelae on 5STS time?

Gait speed performed two trials, with the faster time recorded. How many times did the 5STS test perform and what value did it select?

There are many abbreviations as a whole. Authors should avoid heavy use of abbreviations other than common terms.

Author Response

  • Comment #1: The aim of this study is to analyze the responsiveness of the 5STS test among stroke patients and to estimate the MCIDs for different severity levels of community ambulation and stages of recovery. The reviewer considers that this study to be clinically important, with appropriate study design and data analysis. However, there are some questions in this study.

Response: We would like to thank Reviewer 2 for their consideration and all the comments made.

  • Comment #2: As a comment, as the authors also describe in the “Introduction” session, the 5STS test affects the prognosis and movement disorders of stroke, but it is only one of many aspects. It is not possible to predict the prognosis or movement disorders of stroke with the 5STS test alone. Therefore, reviewer believes that combining the results of other performance tests will make the data even more implicative.

Response: Thank you very much for your comment. Your suggestion suggests a prognostic research, but this is not the purpose of this methodological research study for analyzing responsiveness of the 5STS test. Therefore, other factors outside the 5STS cannot be taken into account, since from the clinimetric analysis approach.

  • Comment #3: The information of stroke patients is poor in this study. The authors should add patients’ background to the “Materials and Methods” session. Particularly, what was the breakdown of the pathosis of 111 stroke patients? (e.g., cerebral infarction, cerebral hemorrhage and subarachnoid hemorrhage). Moreover, how do these differences in pathosis affect 5STS time?

Response: Thank you very much for your comment. This information is described in table 1 of results.

  • Comment #4: Additionally, the reviewer believes that 5STS time is strongly affected by the differences in sequelae of stroke (e.g., hemiplegia and movement disorders). How do the authors interpret the effects of differences in these sequelae on 5STS time?

Response: Thank you very much for your question. We appreciate you asking us this because this is why we have established different subgroups. Previously, different subgroups had not been taken into account, reflected in our study by walking category, so the sequelae and severity levels and their influence on the 5STS times are reflected in the differences that exist for the household and limited subgroups community.

  • Comment #5: Gait speed performed two trials, with the faster time recorded. How many times did the 5STS test perform and what value did it select?

Response: Thank you very much for your question. They had one try. As this test is highly demanding, this test is performed in a trial (Mentiplay et al., 2020. This has been specified in the manuscript.

  • Comment #6: There are many abbreviations as a whole. Authors should avoid heavy use of abbreviations other than common terms.

Response: Thank you very much for your comment. We have eliminated abbreviations limiting ourselves to those commonly used for tests and statistical analysis.

Reviewer 3 Report

The authors took up a difficult topic to analyze the responsiveness of the 5STS test among stroke patients and to estimate the MCIDs for different severity levels of community ambulation and stages of recovery. Despite the fact that the study is very meticulously described and provides many details, e.g. statistical analysis, minimal clinically important differences between tests, personally on the side of the reader is rather difficult to read. The tables mainly contain data about points differences/changes scores. In my opinion the topic has a little importnace for the practitioners. The inclusion criteria like >30 years of age make that there is a wide variety of the study group. In my opinion for patients after stroke this criterium is not objective and should be more precise. The authors also didn’t take into account in the inclusion critaria the type of the stroke. In the conclusions of the study there arn’t any that could be translate into practice. The authors use only the test results, however, they do not define a clear translation into clinical use. I greatly appreciate the Authors' contribution to this difficult work, but I think that this topic doesn’t fit the subject of the journal IJERPH.

Author Response

  • Comment #1: The authors took up a difficult topic to analyze the responsiveness of the 5STS test among stroke patients and to estimate the MCIDs for different severity levels of community ambulation and stages of recovery. Despite the fact that the study is very meticulously described and provides many details, e.g. statistical analysis, minimal clinically important differences between tests, personally on the side of the reader is rather difficult to read. The tables mainly contain data about points differences/changes scores. In my opinion the topic has a little importnace for the practitioners.
  • I greatly appreciate the Authors' contribution to this difficult work, but I think that this topic doesn’t fit the subject of the journal IJERPH.
  • In the conclusions of the study there arn’t any that could be translate into practice. The authors use only the test results, however, they do not define a clear translation into clinical use.
  • The inclusion criteria like >30 years of age make that there is a wide variety of the study group. In my opinion for patients after stroke this criterium is not objective and should be more precise. The authors also didn’t take into account in the inclusion critaria the type of the stroke.

Response: Thank you very much for your comments.

Comment (1a): we want indicate that this manuscript is submitted for the Speccial Issue "Clinimetric Assessment of Instruments for the Measurement of Functional Health Status in Special Populations", so we believe the theme fits perfectly with the IJERPH journal.

Comment (1b):  Second, we have established the clinical use of our results. We say in introduction "This paper focuses on these voids and provides MCIDs which may be used by clinicians and researchers. Clinicians can use this information to interpret the relevance of changes observed in a patient; whereas researchers can use it to define the boundary between change or no change among two groups (eg, those treated with different interventions) and for the calculation of sample sizes. " With our study, we are contributing these MCIDs for 5STS. Furthermore, as we described in the conclusion, "The MCIDs for gait speed also showed that these were influenced by stages of recovery, with more conservative values ​​in the initial stages. We confirmed that clinicians and researchers need to consider stages of recovery and levels of severity when using MCIDs for stroke patients. "Therefore, although we respect that you do not consider our results to be of clinical importance, we do not agree that these are not described in the manuscript or that our results have no clinical applicability.

Comment (1c): in relation to age as an inclusion criterion, variety in age is something commonly used in research about stroke patients. For example, previous studies such as that of Pardo et al. 2013 or Mentiplay et al. 2020 use ≥18 or ≥21 years old, respectively. 

Round 2

Reviewer 2 Report

I think all responses or comments to our questions have been addressed satisfactorily.

I have no comments on the revised manuscript.

Reviewer 3 Report

I can accept manuscript in present form.